# ACTION-CONSTRAINED IMITATION LEARNING

## ABSTRACT

Policy learning under action constraints plays a central role in ensuring safe behaviors in various robot control and resource allocation applications. In this paper, we study a new problem setting termed Action-Constrained Imitation Learning (ACIL), where an action-constrained imitator aims to learn from a demonstrative expert with larger action space. The fundamental challenge of ACIL lies in the unavoidable mismatch of occupancy measure between the expert and the imitator caused by the action constraints. We tackle this mismatch through *trajectory alignment* and propose DTWIL, which replaces the original expert demonstrations with a surrogate dataset that follows similar state trajectories while adhering to the action constraints. Specifically, we recast trajectory alignment as a planning problem and solve it via Model Predictive Control, which aligns the surrogate trajectories with the expert trajectories based on the Dynamic Time Warping (DTW) distance. Through extensive experiments, we demonstrate that learning from the dataset generated by DTWIL significantly enhances performance across multiple robot control tasks and outperforms various benchmark imitation learning algorithms in terms of sample efficiency.

## 1 INTRODUCTION

Reinforcement learning (RL) is commonly used to solve tasks by finding a policy that maximizes cumulative rewards through interactions with the environment. However, in many real-world applications, designing an effective reward function that consistently encourages the desired behavior in all situations is a significant challenge. In such cases, imitation learning (IL) offers a compelling alternative. Rather than relying on a reward function, IL learns a policy directly from a set of pre-collected expert demonstrations, which are transition data logged from a near-optimal policy (Pomerleau & A, 1991; Ho & Ermon, 2016).

In many real-world tasks, ensuring the safe and proper functioning of agents is crucial. To achieve this, we can impose constraints that define the feasible set of actions for the agents. Classic examples include optimally allocating network resources under capacity constraints (Xu et al., 2018; Gu et al., 2019; Zhang et al., 2020) and robot control under kinematic limitations that prevent damage to the robot's physical structure (Pham et al., 2018b; Gu et al., 2017; Jaillet & Porta, 2012; Tsounis et al., 2020). Additionally, in many IL scenarios, the performance gap between the expert and the imitator must be considered. For example, if data is collected using a human to perform tasks, the imitator, which may be a robot with hardware limitations, is likely to be unable to replicate the large-scale human actions. In this case, action constraints are essential to ensure the imitator can safely perform tasks within its own capabilities while still learning from the expert's behavior. While there has been substantial research on action-constrained reinforcement learning (ACRL) (Kasaura et al., 2023; Lin et al., 2021; Brahmanage et al., 2023; Chen et al., 2024), surprisingly, little attention has been given to action-constrained imitation learning (ACIL).

To ensure that the actions generated by the policy adhere to specific constraints during both training and evaluation, most existing ACRL methods incorporate a projection layer on top of the policy network (Chow et al., 2018; Liu et al., 2020; Gu et al., 2017). However, such an approach can cause issues in IL. Most IL approaches aim to minimize the discrepancy between the occupancy measure of the expert demonstrations and that of the imitator (Pomerleau & A, 1991; Ho & Ermon, 2016). When expert actions lie outside the feasible action set, the projection layer can prevent the imitator from accurately matching the occupancy measure of the expert, especially in cases with

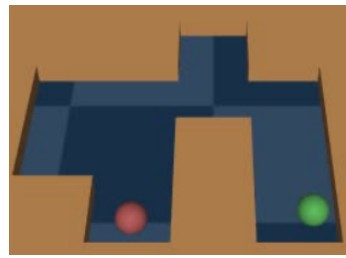 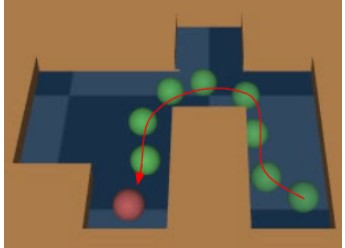 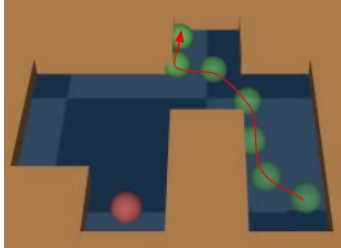

(a) Starting point        (b) Unconstrained case        (c) Action-constrained case

Figure 1: (a) The green sphere starts in the bottom-right corner and navigates toward the red sphere (goal). (b) A policy trained via BC successfully executes a U-turn to reach the target. (c) However, when the box constraint is applied by projection, the sphere struggles to make the sharp U-turn and ends up colliding with the wall.

more restrictive action sets. This issue leads to ambiguity in distribution matching for IL methods under action constraints, a problem we term "occupancy measure distortion."

To better illustrate the issue of occupancy measure distortion, let's consider a simple example of a Maze2d goal-reaching task, as shown in Fig 1. (a). In this task, the green sphere (agent) needs to navigate towards the red sphere (goal), using a two-dimensional action space that controls the force applied along the x- and y-axes. An unconstrained policy trained by behavior cloning (BC)(Pomerleau & A, 1991), based on five expert trajectories, can successfully turn left, avoid colliding with the walls, and reach the goal (Fig 1. (b)). Now, consider a weaker agent with a smaller feasible action set, where a projection layer is applied to its policy network. This weaker agent lacks the force to turn as quickly as the unconstrained agent, resulting in a collision with the wall of the space we carved out (Fig 1. (c)) and getting stuck. This example demonstrates how occupancy measure distortion prevents the agent from accurately replicating the expert's trajectory. Without following the expert's path, the action-constrained agent suffers from the distribution shift, and even encounters unexpected dangers in the environment.

Another approach to preventing learning infeasible actions is to focus on matching the state distribution rather than the state-action distribution of expert demonstrations, a scenario known as Learning from observation (LfO). However, they cannot fully avoid issues related to mismatched state distributions, especially with constrained actions, and they typically require a substantial amount of interaction data with the environment.

The most effective way to eliminate occupancy measure distortion is to ensure that both the expert demonstrations and the learner share the same feasible action set, as this would prevent any distortion from occurring. To accomplish this, we recast trajectory alignment as a planning problem, aiming to generate trajectories that closely resemble the original expert trajectories but consist of constrained actions as surrogate expert demonstrations. We leverage Model Predictive Control (MPC) (Richalet et al., 1978) due to its flexibility in defining objective functions and its compatibility with various constraints. Unlike existing MPC approaches, which primarily focus on optimizing short-horizon returns during planning, we optimize for the similarity between the rollout trajectories and the expert trajectories. To quantify this similarity, we employ Dynamic Time Warping (DTW) (Hiroaki & Chiba, 1978), which allows us to compare trajectories that have different pacing of behaviors. In this paper, we introduce Dynamic Time Warping Imitation Learning (DTWIL), an algorithm designed to generate surrogate action-constrained demonstrations and learn the corresponding policy. Our experiments demonstrate that DTWIL outperforms a range of benchmark IL algorithms in navigation and locomotion tasks, particularly in terms of sample efficiency, while being less susceptible to the challenges posed by occupancy measure distortion.

## 2 RELATED WORK

**Action constrained Reinforcement Learning** To the best of our knowledge, no prior work has specifically addressed the problem of ACIL, which tackles the capability gap between the expert and the learner agent. Therefore, we refer to ACRL methods to define the problem setting in this paper.

Kasaura et al. (2023) provides a benchmark for evaluating existing ACRL approaches. Some works, such as Pham et al. (2018a); Bhatia et al. (2019); Dalal et al. (2018), ensure safe and compliant behavior by incorporating a differentiable projection layer at the end of the policy network to meet action constraints. However, Lin et al. (2021); Brahmanage et al. (2023) highlight issues with this approach, particularly the zero gradient and longer training times, and propose alternative methods. Notably, Brahmanage et al. (2023); Chen et al. (2024) employ normalizing flows to directly generate actions that comply with the constraints, thereby circumventing the drawbacks associated with projection layers.

**Learning from Demonstration**  IL focuses on deriving a policy using only the information from expert demonstrations, which also termed Learning from Demonstration (LfD). BC (Pomerleau & A, 1991) approaches this by treating policy as a state-action mapping, learning it in a supervised manner. Adversarial Imitation Learning (AIL), on the other hand, focuses on matching the state-action distribution between expert and learner through adversarial training. GAIL (Ho & Ermon, 2016) is a foundational method in this domain, using a discriminator to distinguish between expert and learner transitions, and providing rewards based on this discrimination. Various AIL extensions (Kostrikov et al., 2019a;b) improve on GAIL, tailoring the method to different environments and goals. A comprehensive review of IL techniques can be found in Zare et al. (2024), but ACIL remains unexplored in these surveys.

**Learning from Observation**  An alternative approach to avoid the undesirable effects of projected policy outputs after imitating expert actions is to learn from expert observation data only, which falls under the scenario of Learning from Observation (LfO). Methods like GAIfO and IDDM (Torabi et al., 2018b; Yang et al., 2019) follow the principles of GAIL by training a state-only discriminator. OPOLO (Zhu et al., 2020) further improves on this by relaxing the on-policy requirement, speeding up the learning process. BCO (Torabi et al., 2018a) takes a different approach by learning an inverse dynamics model to infer the expert's missing actions from observations, and then applying BC to train the policy. CFIL (Freund et al., 2023), using a flow-based model to capture state or state-action distributions, sets a new benchmark for LfO scenario. However, despite relying solely on expert state information, these methods still overlook the capability gap between the expert and the learner agent, and many of them depend on a large amount of environment interaction data.

**Cross-Embodiment Imitation Learning**  Cross-Embodiment Imitation Learning focuses on transferring knowledge or skills between agents with different physical structures, such as robots with varying morphologies or dynamics. This field addresses the challenges of aligning state and action spaces across embodiments to enable effective knowledge transfer. Approaches in this domain often leverage shared latent spaces, domain adaptation techniques, or hierarchical reinforcement learning to bridge embodiment-specific differences. For example, modular policy frameworks (Huang et al., 2020) and domain randomization strategies (Tobin et al., 2017) have been employed to achieve generalization across multiple embodiments. While ACIL also seeks to address the challenge of transferring knowledge across different agents, it does not consider differences in physical structures. Instead, ACIL focuses on a unique problem setting where agents share action spaces of the same dimension but differ in the scale or magnitude of their actions.

## 3 PRELIMINARIES

**Problem Formulation**  We consider a Markov decision process (MDP) defined as a tuple $\mathcal{M} = \langle \mathcal{S}, \mathcal{A}, T, r, p_0, \gamma \rangle$, where $\mathcal{S}$ and $\mathcal{A}$ are the sets of feasible state and action respectively; $T$ describes the dynamics of the environments, with $T(s_{t+1}|s_t, a_t)$ indicating the transition probability to next state $s_{t+1}$ from the current state $s_t$ if the agent takes action $a_t$; $p_0$ is the initial state distribution; $R : \mathcal{S} \times \mathcal{A} \to \mathbb{R}$ is the reward function; and $\gamma \in [0, 1]$ is the discount factor. An agent follows its policy $\pi : \mathcal{S} \to \mathcal{A}$ to interact with the environment of MDP with an objective of maximizing long-term expected cumulative reward. In this paper, we consider action-constrained MDPs where for each state $s \in \mathcal{S}$ there is a feasible action set $\mathcal{C}(s) \subseteq \mathcal{A}$ determined by explicit action constraints incorporated. That is, the agent can only take actions from $\mathcal{C}(s)$ at each time step.

**Model Predictive Control**  In actor-critic RL, solving an MDP is to find the optimal policy $\pi^*$ maximizing cumulative reward. In control, the optimal policy is formulated by maximizing a spe-

cific performance measure. MPC achieves this by utilizing a forward dynamics model $f(s_t, a_t)$ of the environment to explore various action sequences. This allows MPC to evaluate potential future trajectories and select the one that best meets the defined objective $J$. A local solution to the trajectory optimization at each step $t$ can be acquired by estimating the optimal action sequence $a_{t:t+H}$ over a finite horizon $H$:

$$\pi_{\text{MPC}}(s_t) = \arg\min_{a_{t:t+H}} \mathbb{E}\left[\sum_{i=t}^{H} J(s_i, a_i)\right], \tag{1}$$

The agent will execute the first action of the resulting action sequence, and repeat the procedure again at the next time step. To improve action sampling, we can utilize the Cross-Entropy Method (CEM) optimizer, which iteratively refines the mean ($\mu$) and standard deviation ($\sigma$) of a multivariate Gaussian distribution by sampling actions, evaluating them, and updating the distribution based on the best samples over a finite horizon. In this work, we employ an MPC implementation based on Probabilistic Ensembles with Trajectory Sampling (PETS) as proposed by Chua et al. (2018). PETS integrates probabilistic neural networks to model the dynamics of the environment, utilizing an ensemble of learned models to estimate uncertainty in predictions. This ensemble approach allows for more robust decision-making by accounting for variability in the system. In practice, PETS interacts with the environment by iteratively predicting future states based on the current state, choosing actions that maximize a given reward function while considering uncertainty, and then updating its models as new data is collected. This method significantly reduces the sample complexity, allowing the agent to perform well after a limited number of interactions with the environment.

**Dynamic Time Warping**   DTW (Hiroaki & Chiba, 1978) is an algorithm designed to measure the similarity between two temporal series data that may not align perfectly in time. It is particularly useful in scenarios where trajectories, such as those generated by agents with different action constraints, differ in speed or timing but represent the same underlying behavior. The core of DTW lies in the calculation of the optimal warping path $\rho^*$ and the resulting DTW distance, which quantifies the alignment cost. Specifically, let $\mathbf{x} = \{x_1, x_2, \ldots, x_n\}$ and $\mathbf{y} = \{y_1, y_2, \ldots, y_m\}$ denote two sequences of length $n$ and $m$, respectively, then the DTW distance between $\mathbf{x}$ and $\mathbf{y}$ is given by

$$\text{DTW Distance}(\mathbf{x}, \mathbf{y}) = \sum_{(i,j)\in\rho^*} \|x_i - y_j\|^2 = \min_{\rho} \sum_{(i,j)\in\rho} \|x_i - y_j\|^2,$$

where $\rho = \{(i_k, j_k)\}_{k=1}^{K}$ is a warping path such that:

1. $i_1 = 1$ and $j_1 = 1$,
2. $i_K = n$ and $j_K = m$,
3. $i_k \leq i_{k+1}$ and $j_k \leq j_{k+1}$ for all $k$,
4. $|i_{k+1} - i_k| \leq 1$ and $|j_{k+1} - j_k| \leq 1$ for all $k$.

---

**Algorithm 1** Dynamic Time Warping Imitation Learning (DTWIL)

---

1: **Input:** Expert demos $\tau = \{\tau^i\}_{i=1}^N$, planning horizon $H$, ERC horizon $h_{\text{erc}}$, number of particles $P$, dynamics model ensembles $f$, training dataset $\mathcal{D} = \{\tau^i\}_{i=1}^N$, the number of episodes to run $K$
2: BC dataset $\mathcal{D}_{\text{BC}} \leftarrow \{\}$
3: **for** Iteration $k = 1$ to $K$ **do**
4:     Select an expert trajectory $\tau^i$
5:     Train $f$ with $\mathcal{D}$
6:     $\tau^{c_i} \leftarrow$ Trajectory Alignment($\tau^i$)
7:     $\mathcal{D} \leftarrow \mathcal{D} \cup \tau^{c_i}$
8:     **if** no alignment of $\tau^i$ in $\mathcal{D}_{BC}$ **or** DTWDistance($\tau^{c_i}, \tau^i$) < DTWDistance($\mathcal{D}_{\text{BC}}[i], \tau^i$) **then**
9:         $\mathcal{D}_{\text{BC}}[i] \leftarrow \tau^{c_i}$
10:    **end if**
11: **end for**
12: Train a BC policy with $\mathcal{D}_{\text{BC}}$

---

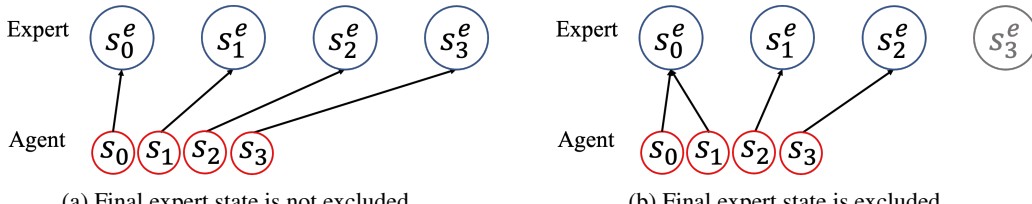

(a) Final expert state is not excluded

(b) Final expert state is excluded

Figure 2: Effect of excluding the final expert state on the DTW warping path. Including the final expert state Figure 2a leads to a 1-to-1 alignment since both trajectories have the same number of states. Excluding it Figure 2b prevents state from advancing, yielding a more desirable matching. The total arrow length represents the DTW distance.

## 4 METHODOLOGY

Our motivation is to generate a surrogate demonstration dataset that aligns with expert trajectories while operating within constrained action spaces , and later utilize this surrogate data set to train a BC policy for generalization. To this end, we recast the alignment issue as a trajectory planning task, where a trajectory of the agent is designed to follow the expert demonstration. As mentioned in Section 3, we leverage the PETS framework (Chua et al., 2018) to optimize the expected outcomes of sampled actions. In this process, we replace the environment reward with DTW (Hiroaki & Chiba, 1978) distance as our key criterion for selecting actions, ensuring better alignment with the expert trajectory. Additionally, to handle the complexities of environments requiring precise movements, we introduce Expert Regularized Control (ERC), inspired by Actor Regularized Control (ARC) (Sikchi et al., 2021), into the trajectory sampling process, improving the alignment's effectiveness.

In the following sections, we detail our implementation of DTW distance as the action selection criterion in Section 4.1, highlighting its role in aligning the agent's trajectory with that of the expert. Section 4.2 introduces ERC and its integration into the trajectory sampling process. The comprehensive pseudo code for DTWIL can be found in Algorithm 1, and the pseudo code for trajectory alignment is presented in Algorithm 2, and

### 4.1 TRAJECTORY ALIGNMENT

Due to the asynchronous nature of the rollout pacing between the expert demonstration and the constrained agent, step-by-step alignment is not feasible. To address this, we incorporate DTW to evaluate the alignment and select the most appropriate planning trajectory that corresponds to the expert demonstration. In the following sections, we explain how DTW distance is utilized as a criterion for the MPC controller in PETS framework in Section 4.1.1 and how we determine the expert demonstration segment to be aligned at each step in Section 4.1.2.

#### 4.1.1 DTW CRITERIA

To utilize DTW as a reference, we first introduce a progression parameter, $t_{\text{pg}}$, which indicates the timestep of the expert state with which the constrained agent is currently aligned. For instance, if the current progress is at $t_{\text{pg}}$, and the planning horizon is set to $H$, the targeted segment of the expert trajectory for alignment would be $s^{\text{e}}_{t_{\text{pg}}:(t_{\text{pg}}+H)}$, where $s^{\text{e}}_t$ denotes the $t$-th expert state.

Let the current timestep be $t$, the current progress be $t_{\text{pg}}$, and the $H$-step planning trajectory rolled out by the action sequence $A$ and a dynamics model $f_\theta$ be $s_{t:(t+H)}$. The optimal planning action sequence $A^*$ is then defined as:

$$A^* = \arg\min_A \ \mathbb{E}\left[\text{DTWDistance}\big(s^{\text{e}}_{t_{\text{pg}}:(t_{\text{pg}}+H)}, s_{t:(t+H)}\big)\right]. \tag{2}$$

We approximate the solution to the optimization problem by employing a CEM optimizer, which samples 500 candidate action sequences and selects the one with the smallest DTW distance to the expert trajectory. To address variations in scale across different dimensions, we normalize both

---

**Algorithm 2** Trajectory Alignment

---

1: **Input:** Planning horizon $H$, ERC horizon $h_{\text{erc}}$, number of particles $P$, dynamics model ensembles $f$, $i$-th expert trajectory $\tau^i = \{(s^{e_i}, a^{e_i})\}_{t=0}^l$, constrained action space $\mathcal{C}(s)$.
2: **Output:** $\tau^{c_i}$
3: Agent's initial state $s_0 \leftarrow s_0^{e_i}$, progression $t_{\text{pg}} \leftarrow 0$, time step $t \leftarrow 0$, alignment $\tau^{c_i} \leftarrow \{\}$
4: Action projection function Proj()
5: **while** $t < \text{max\_episode\_steps}$ **and** $t_{\text{pg}} < l$ **do**
6:     **if** $t_{\text{pg}} + H > l$ **then**
7:         Pad the target expert segment to length$= H$ with $s_l^{e_i}$.
8:     **end if**
9:     **for** Particle $p = 1$ to $P$ **do**
10:         **for** Action sampled $a_{t+h}^p$ from CEM, $h = 0$ to $H$ **do**
11:             **if** $h \leq h_{\text{erc}}$ **then**
12:                 $a_{t+h}^p \leftarrow \beta \, \text{Proj}\big( a_{\min(t_{\text{pg}}+h,l)}^{e_i} \,|\, \mathcal{C}(s_{\min(t+h,l)}^p) \big) + (1-\beta) \, a_{t+h}^p$
13:             **end if**
14:             $s_{t+h+1}^p = f(s_{t+h}^p | a_{t+h}^p)$
15:         **end for**
16:         $\|p\|_{\text{DTW}} \leftarrow \text{DTWDistance}(s_{t:t+H}^p, \; s_{(t_{\text{pg}}):(t_{\text{pg}}+H)}^{e_i})$
17:     **end for**
18:     $p^* \leftarrow \arg\min_p \|p\|_{\text{DTW}}$
19:     Update CEM distribution
20:     Execute $a_t^{p^*}$ and get $s_{t+1}$
21:     $\tau^{c_i} \leftarrow \tau^{c_i} \cup (s_t, a_t^{p^*})$
22:     **if** Progression has advanced in the warping path **then**
23:         $t_{\text{pg}} \leftarrow t_{\text{pg}} + 1$
24:     **end if**
25: **end while**

---

the planned trajectory and the corresponding expert trajectory segment prior to computing the DTW distance. Specifically, each dimension is linearly scaled such that the minimum and maximum values of the expert trajectories are mapped to 0 and 1, respectively. To ensure compatibility with the action-constrained setting, we adapt the CEM optimizer through rejection sampling, strictly enforcing that all sampled actions satisfy the imposed constraints. Subsequently, the MPC controller executes the first action of $A^*$.

### 4.1.2 PROGRESSION MANAGEMENT

The progression parameter, $t_{\text{pg}}$, is initialized to 0 at the start of every trajectory alignment. After each action, we update $t_{\text{pg}}$ by analyzing the warping map to determine how many expert states the agent's action has advanced. Notably, when constructing the warping path, the final expert state in the segment is excluded from the matching calculation to prevent unintended progression when the agent exhibits minimal movement across consecutive actions. Specifically, when two trajectories have an equal number of states, DTW often tends to align states in a strictly 1-to-1 manner, which can mislead progression. By excluding the final expert state, the DTW algorithm is encouraged to create a 2-to-1 alignment during the matching process. Given the constrained actions, which naturally take smaller steps than expert actions, this 2-to-1 alignment often occurs in the initial few states. Consequently, if the agent's first planning state, $s_1$, is not sufficiently close to the next expert state, $s_1^e$, it is more likely to be matched with the current expert state, $s_0^e$. This concept is illustrated in Figure 2.

Figure 3 shows how this advancement value is determined. The advancement value is then added to $t_{\text{pg}}$ after every MPC step.

### 4.2 EXPERT REGULARIZED CONTROL

In environments that demand precise movements, even small errors can lead to significant disruptions. To mitigate this, we incorporate expert actions into the sampled actions as guidance , termed

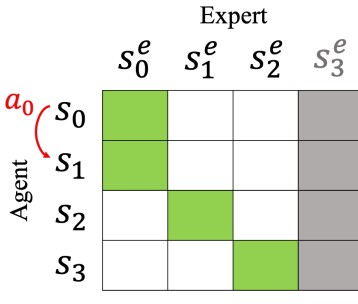

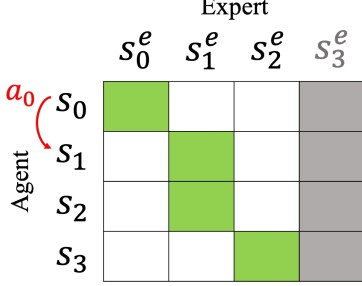

(a) Progression Advancement = 0      (b) Progression Advancement = 1

Figure 3: Since the MPC controller executes only the first planning step per iteration, we focus on the number of expert states the agent advances after the initial action $a_0$. The figure shows two DTW warping path cases (green patches). In Figure 3a, the agent transitions from $s_0$ to $s_1$ while staying aligned with $s_0^e$ causing no progression ($t_{\text{pg}}$ unchanged). In Figure 3b, the agent advances to the next expert state, updating $t_{\text{pg}}$ to $t_{\text{pg}} + 1$.

**ERC.** Specifically, the actions used to rollout the planning trajectories in the MPC controller become the weighted average of the sampled actions and a corresponding segment of the expert demonstration. To implement this, we first extract a specific segment $a_{t_{\text{pg}}:(t_{\text{pg}}+h_{\text{erc}})}^{\text{e}}$, from the expert actions $a^{\text{e}}$, where $h_{\text{erc}}$ is the horizon over which expert actions are blended. Then, given the dynamics model ensembles $f(s, a)$, a specific weight $\beta \in [0, 1]$, and the projection function $\text{Proj}(a \,|\, \mathcal{C}(s))$, which projects an action $a$ onto a specific constrained action space $\mathcal{C}(s)$, ERC guide the trajectory generation with the following functions:

$$\text{For } h = 0, 1, .., H :$$
$$a_h = \begin{cases} \beta \, \text{Proj}(a_{t_{\text{pg}}+h}^{\text{e}} \,|\, s_h) + (1 - \beta) \, a_h^{\text{sampled}}, & \text{if } h <= h_{\text{erc}}, \\ a_h^{\text{sampled}}, & \text{if } h > h_{\text{erc}}, \end{cases} \tag{3}$$
$$s_{h+1} = f(s_h, a_h),$$

where $a_h$ is the $h^{\text{th}}$ action step in an $H$-step planning trajectory, $a_h^{\text{sampled}}$ is the $h^{\text{th}}$ action directly sampled from a CEM optimizer, and $s_h$ is the $h^{\text{th}}$ state of the planning trajectory.

The performance of our algorithm in environments where agents are highly susceptible to deviations—such as Hopper, where falling results in early termination—is significantly enhanced by incorporating ERC. A detailed analysis of this improvement is presented in Section 5.6.

## 5 EXPERIMENTS

In this chapter, we assess DTWIL across a range of randomly initialized continuous control tasks in navigation and locomotion environments, each subject to different constraints. We compare our results against both offline baselines and online baselines. For a fair comparison, we allocate the same number of environment steps to the online baselines as we do to DTWIL.

Two types of constraints are applied: box constraints and state-dependent constraints. A box constraint, denoted as $\text{Box}(c_{\text{box}})$, restricts each action dimension to the range $[-c_{\text{box}}, c_{\text{box}}]$, where $c_{\text{box}}$ is a positive constant. In contrast, a state-dependent constraint varies based on the agent's current state. To ensure that these baseline methods adhere to the constrained action domains, we project their generated actions onto the nearest feasible actions based on the $L_2$ norm.

### 5.1 CONSTRAINED ENVIRONMENTS

**Maze2d (Fu et al., 2020)** To evaluate our method on a navigation task, we selected the Maze2d-Medium-v1 environment. This task involves a point-mass agent navigating a 2D maze from a randomly chosen start location to a goal. The original action set is a 2-dimensional vector $(v_1, v_2)$ with

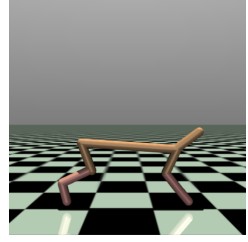 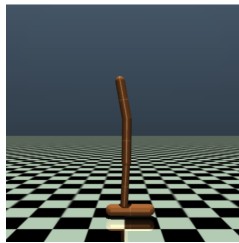

Figure 4: We evaluate the impact of action constraints on DTWIL and baseline methods across three environments : Maze2d-Medium-v1, HalfCheetah-v3, and Hopper-v2.

each element in the range $[-1.0, 1.0]$. We impose an Box$(0.1)$ constraint and a state-dependent constraint **M+O** defined as $\Sigma_{i=1}^{2}|v_i w_i| \leq 0.5$ on agent actions, where $(w_1, w_2)$ represent the velocities in the x and y directions, respectively. For this task, we collected 100 demonstrations, resulting in a total of 18,525 state-action pairs for training.

**HalfCheetah (Brockman et al., 2016)**    The task involves controlling a bipedal cheetah agent to run forward by applying torque to its joints. The action space consists of a 6-dimensional vector $(v_1, v_2, ..., v_6)$, where each component is bounded by $[-1, 1]$. We introduce a Box$(0.5)$ constraint and a state-dependent constraint **HC+O** defined as $\Sigma_{i=1}^{6}|v_i w_i| \leq 10$, where $w_i$ denotes the angular velocity of the $i$-th joint, a component of the agent's state. We rely on five 1000-step expert demonstrations for training.

**Hopper (Brockman et al., 2016)**    The task requires controlling a robot to hop forward by applying torques to its hinges. The action is represented by a 3-dimensional vector $(v_1, v_2, v_3)$, with each value constrained between $[-1.0, 1.0]$. We also impose two separate constraints on this task. The first one is a Box$(0.9)$ constraint, while the second introduces a state-dependent constraint **H+M**: $\Sigma_{i=1}^{3}|v_i w_i| \leq 10$ , where $w_i$ denotes the angular velocity of the $i$-th joint, which is part of the robot's state. For training, we use five expert demonstrations, each consisting of 1000 state-action pairs.

## 5.2    BASELINES

To ensure that the action outputs of various baseline methods meet specific constraints, we incorporate a projection layer into each method's policy, allowing the action outputs to remain within the feasible set. We append "+P" to the names of each baseline method to denote the versions of the algorithms that include a projection layer.

- **BC+P (Pomerleau & A, 1991):** BC formulates policy learning as a supervised problem, treating the policy as a mapping between states and actions.

- **BCO+P (Torabi et al., 2018a):** BCO is a LfO method, learning an inverse dynamics model to infer action from state-only data and applying BC to learn a policy.

- **GAIL+P (Ho & Ermon, 2016):** GAIL is an online LfD method that utilize a generative adversarial network (GAN) to infer the underlying reward function.

- **GAIfO+P (Torabi et al., 2018b):** Similar to GAIL but only learning from observations, GAIfO is an AIL-based online LfO algorithm.

- **OPOLO+P (Zhu et al., 2020):** OPOLO is an online LfO method. Leveraging off-policy learning, OPOLO ranks among the most effective LfO techniques.

- **CFIL-s+P/ CFIL-sa+P (Freund et al., 2023):** CFIL utilize a flow-based model to capture state or state-action distributions, sets a new benchmark for LfO scenario. The LfD version of CFIL is denoted as CFIL-sa, and LfO version of CFIL is denoted as CFIL-s.

## 5.3 PERFORMANCE COMPARISON

In all tasks, DTWIL only interacts with the environment using MPC for no more than 50K steps. To ensure a fair comparison, we limit the interaction for all online IL methods to 50K environment steps during training. All results are evaluated with randomly initialized starting states.

Following this, the best-performing model from each algorithm during these interactions was selected for final evaluation. This ensures that the results reflect the effectiveness of each method within a limited sample regime, providing a fair comparison across environments while emphasizing sample efficiency.

| Task | Maze2d box | Maze2d M+O | HalfCheetah Box | HalfCheetah HC+O | Hopper Box | Hopper H+M |
|---|---|---|---|---|---|---|
| BC+P | 0.61 ± 0.05 | 0.81 ± 0.05 | 1815.51 ± 303.89 | **2753.86 ± 27.34** | 2204.83 ± 753.32 | 1233.96 ± 211.87 |
| GAIL+P | 0.22 ± 0.0 | 0.14 ± 0.05 | -163.63 ± 47.47 | -185.53 ± 66.11 | 360.97 ± 59.19 | 261.83 ± 81.41 |
| BCO+P | 0.14 ± 0.05 | **0.88 ± 0.06** | -4.05 ± 4.07 | 6.23 ± 31.85 | 219.46 ± 20.33 | 224.25 ± 32.81 |
| GAIfO+P | 0.07 ± 0.02 | 0.19 ± 0.08 | -74.77 ± 32.98 | -163.84 ± 33.79 | 197.36 ± 30.12 | 206.37 ± 19.19 |
| OPOLO+P | 0.2 ± 0.06 | 0.64 ± 0.13 | -605.84 ± 390.21 | -9.12 ± 80.47 | 1068.3 ± 952.96 | 228.28 ± 33.10 |
| CFIL-sa+P | 0.23 ± 0.21 | 0.47 ± 0.10 | -95.67 ± 515.43 | 1674.75 ± 1316.81 | 1485.74 ± 677.37 | 1553.86 ± 1096.28 |
| CFIL-s+P | 0.23 ± 0.06 | 0.45 ± 0.12 | -172.56 ± 738.44 | 1422.98 ± 1830.51 | 866.27 ± 249.20 | 1443.06 ± 547.59 |
| **DTWIL** | **0.77 ± 0.04** | 0.87 ± 0.04 | **2669.41 ± 4.56** | 2637.34 ± 26.82 | **2844.68 ± 57.77** | **2873.88 ± 240.46** |

Table 1: Evaluation performance of the proposed method and baseline algorithms across various tasks, with results expressed as the mean and standard deviation calculated from three seeds.

Based on the experimental results, the BC+P algorithm maintains basic functionality across all tasks but is still affected by action constraints, which hinders its ability to replicate expert-level performance. This limitation is particularly noticeable in the Hopper environment, where a single fall results in the episode ending prematurely, further hindering its performance. The rigid constraints imposed on the actions make it challenging for BC+P to generalize well in tasks requiring smooth and dynamic control.

Moreover, the other online algorithms such as GAIL+P and OPOLO+P face dual challenges. Not only are they affected by the same action constraints, but they also suffer from poor sample efficiency, which leads to subpar performance across all tasks. These methods, despite interacting with the environment, cannot recover expert-like behavior within the limited number of interaction steps, contributing to their consistently low scores. While BCO+P show competitive performance in simpler tasks like Maze2d M+O, they fall short in more complex environments.

In contrast, DTWIL, which learns from surrogate expert data and adopts a BC approach to learn the policy, perform well across all tasks. By learning from the surrogate data to match the expert trajectories and using BC for policy learning, DTWIL manages to replicate expert performance while maintaining sample efficiency. As a result, it successfully reproduces expert-like trajectories across tasks, without being adversely affected by the constraints that cripple other methods. The results of training the various baseline methods for sufficient steps are included in Appendix A.3.

## 5.4 PREVENTION FROM UNINTENDED PROGRESSION

To mitigate unintended progression of the parameter $t_{pg}$, as detailed in 4.1.2, we exclude the terminal state of the alignment target during computation. As demonstrated in Table 2, this adjustment significantly enhances performance in the Maze2d-Medium environment under box constraints. Specifically, excluding the final expert state when determining the DTW warping path improves the returns obtained during both the trajectory alignment phase and the subsequent behavioral cloning (BC) phase. These results validate the effectiveness of the proposed modification in stabilizing and optimizing the alignment process.

| | Excluded | Not Excluded |
|---|---|---|
| DTW-S | 2.99 ± 0.75 | 2.99 ± 0.82 |
| Return-S | **0.76 ± 0.0** | 0.69 ± 0.0 |
| Return-BC | **0.77 ± 0.04** | 0.72 ± 0.03 |

Table 2: Results comparison of whether the final expert state is excluded when calculating the warping path in Maze2d-Medium under the box constraint.

| Task | HalfCheetah Box | HalfCheetah Box-Sync | Hopper Box | Hopper Box-Sync |
|------|-----------------|----------------------|------------|-----------------|
| DTW-S | 15.17 ± 0.24 | **15.06 ± 0.12** | **11.70 ± 6.02** | 27.68 ± 0.26 |
| Return-S | 2576.20 ± 61.62 | **2590.31 ± 24.07** | **2527.63 ± 572.53** | 418.73 ± 89.35 |
| Return-BC | **2669.41 ± 4.56** | 2594.28 ± 29.80 | **2844.68 ± 57.77** | 153.52 ± 1.20 |

Table 3: Comparison of results between asynchronous and synchronous progression methods. DTW-S denotes the DTW distance between the generated surrogate trajectories and the expert trajectories, Return-S indicates the average return of the surrogate expert data, and Return-BC represents the average return of BC policy trained on this surrogate expert data.

## 5.5 ASYNCHRONOUS PROGRESSION UPDATE

In this section, we compare two approaches to progression management. The first is asynchronous progression, where the parameter $t_{pg}$ is updated in tandem with the warping path. This method is primarily used in our algorithm. The second is synchronous progression, where $t_{pg}$ increases by 1 with each step, matching the expert's pace. Given that agents with constrained actions typically take longer to replicate expert behavior, asynchronous progression is more sensible. Table 3 presents the full experimental results for both methods. While the differences on HalfCheetah are minimal, asynchronous progression significantly outperforms on Hopper.

## 5.6 EXPERT REGULARIZED CONTROL

We evaluate the effectiveness of our ERC design in the Hopper environment. Table 4 demonstrate a clear performance difference: without ERC, the agent frequently falls, leading to significantly lower rewards and shorter trajectories. In contrast, incorporating ERC stabilizes the agent's behavior, allowing it to generate surrogate trajectories of appropriate length and maintain consistent performance throughout the task. This highlights the importance of ERC in enabling robust and reliable imitation under action-constrained settings. Refer to Appendix A.5 for detailed hyperparameter tuning.

|  | Without ERC | With ERC |
|--|-------------|----------|
| Return-S | 820.7 ± 84.8 | **2527.6 ± 572.5** |
| Return-BC | 889.7 ± 5.4 | **2844.7 ± 57.8** |

Table 4: Comparison of results with and without ERC applied during action sampling in Hopper.

## 6 CONCLUSION

ACIL has the potential to greatly influence real-world robot training, as real robots often operate under constrained action spaces due to limited power, mechanical imperfections, or restricted capabilities resulting from wear and tear. These limitations present challenges that previous methods have not effectively addressed. In this paper, we highlight that directly learning from expert demonstrations using agents with constrained action spaces introduces several issues, including occupancy measure distortion and asynchronous progression. These challenges cannot be resolved by traditional RL and IL methods because of the inevitable progression gap between expert and agent trajectories. To address this, we propose the first-ever ACIL method, DTWIL, which effectively bridges the gap caused by asynchronous time series alignment. DTWIL leverages DTW distance as a reference to select optimal actions in a MPC framework, and incorporates Actor Regularized Critic (ARC) to stabilize the sampled actions. As a result, our approach outperforms methods heavily reliant on projection in multiple environments, demonstrating that a dedicated algorithm for the ACIL problem is both effective and necessary. Our results indicate that as long as the computational cost of DTW is manageable, DTWIL achieves exceptional performance on ACIL tasks. As the first contribution to the ACIL research field, we hope our work inspires further research. Future efforts could focus on developing ACIL algorithms that handle more complex environments with greater efficiency.

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

# A APPENDIX

## A.1 CEM OPTIMIZER

Our implementation of the CEM optimizer closely follows the approach used in PETS (Chua et al., 2018), where a momentum term is added into the update calculations, and bounds are imposed on the standard deviations in addition to the standard CEM optimization.

Specifically, if a distribution at CEM iteration $i$, $\mathcal{N}(\mu_i, \sigma_i^2)$, is updated toward a target distribution $\mathcal{N}(\mu_{\text{target}}, \sigma_{\text{target}}^2)$, the resulting updated distribution at iteration $i + 1$, $\mathcal{N}(\mu_{i+1}, \sigma_{i+1}^2)$, will be given by:

$$\mathcal{N}(\mu_{i+1}, \sigma_{i+1}^2) = \mathcal{N}(\ \alpha\mu_i + (1-\alpha)\mu_{\text{target}},\ \alpha\sigma_i^2 + (1-\alpha)\sigma_{\text{target}}^2\ ),\ \alpha \in [0, 1], \qquad (4)$$

and the value of $\sigma_i^2$ is further constrained by $\frac{1}{2}w$, where $w$ represents the minimum distance from $\mu_i$ to the bounds of the feasible action space.

Moreover, to adapt the CEM optimizer for our action-constrained setting, we employ rejection sampling to ensure that all sampled actions strictly adhere to the predefined constraints.

## A.2 DYNAMICS MODEL

In this work, we train an ensemble of probabilistic neural networks to model the system's dynamics. Specifically, we utilize ensembles of five dynamics models, where the $b^{th}$ model, $f_{\theta_b}$, is parameterized by $\theta_b$. Each network in the ensemble is trained to minimize the negative log-likelihood of the predicted outcomes, optimizing the following objective:

$$\mathcal{L}(\theta_b) = -\sum_{n=1}^{N} \log f_{\theta_b}(s_{n+1}|s_n, a_n). \qquad (5)$$

Referring to the ensembles used in PETS (Chua et al., 2018), we define our network to output a Gaussian distribution with diagonal covariance parameterized by $\theta$ and conditioned on $s_n$ and $a_n$, i.e.: $f = Pr(s_{t+1}|s_t, a_t) = \mathcal{N}(\mu_\theta(s_t, a_t), \sum_\theta(s_t, a_t))$. In this specific case, Eq. (5) becomes:

$$\mathcal{L}_G(\theta_b) = \sum_{n=1}^{N} [\mu_{\theta_b}(s_n, a_n) - s_{n+1}]^\top \mathbf{\Sigma}_{\theta_b}^{-1}(s_n, a_n) [\mu_{\theta_b}(s_n, a_n) - s_{n+1}] + \log \det \mathbf{\Sigma}_{\theta_b}(s_n, a_n),$$
$$(6)$$

The next states are obtained in the same manner as $\mathbf{TS}\infty$ described in PETS.

Additionally, to mitigate the risk of over-fitting that can occur when a dynamics model is trained solely on expert trajectories, we augment the training data with online agent experiences and iteratively retrain the dynamics models.

## A.3 TRAINING CURVES FOR BASELINE METHODS WITH ADDITIONAL STEPS

In Section 5.3, we presented the performance of DTWIL and various baseline methods when interacting with the environment for up to 50K steps, focusing on sample efficiency. In Figure 5, we showcase the training curves of baseline methods over 500 thousand steps, which is 10 times the original limit. These results reveal that methods like CFIL and OPOLO can train effective policies on multiple tasks when granted sufficient interaction steps. However, compared to DTWIL, which requires only the training of an MPC dynamics model to generate surrogate expert demonstrations, these online LfO methods demand significantly more interaction steps, highlighting their inefficiency relative to DTWIL.

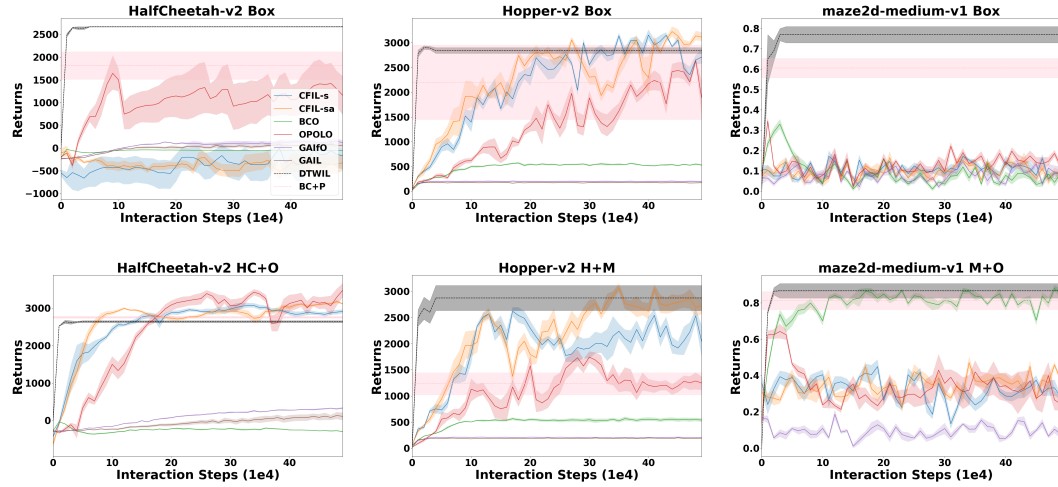

Figure 5: Training curves for baseline methods over 1 million interaction steps across multiple tasks.

## A.4 DTW INPUT NORMALIZATION

Typically, trajectories are normalized before being fed into the DTW calculation, as described in 4.1.1. In this section, we analyze the impact of this normalization. Table 5 shows an ablation study on HalfCheetah and Hopper with their respective box constraints. We observe a performance drop in both environments when this normalization step is omitted from DTWIL. This is because, without normalization, DTW becomes disproportionately influenced by dimensions with larger scales, leading to poor generalization. Conversely, when the states are normalized in advance, DTW treats each dimension equally, resulting in more effective warping.

| Task | HalfCheetah Box | HalfCheetah Box w/o N | Hopper Box | Hopper Box w/o N |
|------|-----------------|------------------------|------------|-------------------|
| Return-S | **2576.2 ± 61.62** | 1667.46 ± 51.13 | **2527.63 ± 572.53** | 608.18 ± 208.20 |
| Return-BC | **2669.41 ± 4.56** | 1893.9 ± 71.56 | **2844.68 ± 57.77** | 281.13 ± 31.88 |

Table 5: Impact of DTW input normalization on performance. Return-S represent the average return of surrogate expert data, while Return-BC denotes the average evaluation return of the BC policy trained on this surrogate data. "W/o N" indicates results obtained without applying DTW input normalization.

| | $\beta = 0$ | $\beta = 0.02$ | $\beta = 0.05$ | $\beta = 0.1$ | $\beta = 0.2$ |
|------|-------------|----------------|----------------|----------------|----------------|
| Return-S | 820.71 ± 84.78 | 1492.97 ± 144.35 | **2527.63 ± 572.53** | 1657.47 ± 286.44 | 670.72 ± 328.28 |
| Return-BC | 889.65 ± 5.39 | 1138.85 ± 56.35 | **2844.68 ± 57.77** | 2167.3 ± 360.73 | 723.95 ± 345.70 |

| | $h_{erc} = 0$ | $h_{erc} = 5$ | $h_{erc} = 10$ | $h_{erc} = 20$ |
|------|---------------|---------------|----------------|----------------|
| Return-S | 820.71 ± 84.78 | **2527.63 ± 572.53** | 2425.25 ± 370.40 | 2166.99 ± 351.04 |
| Return-BC | 889.65 ± 5.39 | **2844.68 ± 57.77** | 2686.85 ± 135.64 | 2616.09 ± 102.90 |

Table 6: Impact of varying $\beta$ and $h_{erc}$ values on performance in the Hopper task with H+M constraints. The table highlights the optimal balance between expert actions and MPC sampling, showing the best-performing configurations for stability and action guidance.

## A.5 HYPERPARAMETERS IN ERC

We explore the influence of the hyperparameter $\beta$, which regulates the balance between expert actions and MPC-sampled actions in the ERC method. Additionally, we examine the effect of the

horizon length $h_{\text{erc}}$, which determines how many steps to blend MPC-sampled actions with expert actions. We conducted experiments on the Hopper with H+M constraints, varying $\beta$ from 0 to 0.2 and $h_{\text{erc}}$ from 0 to 30, while keeping all other hyperparameters fixed at their optimal values identified in prior tuning. As shown in Table 6, setting $\beta$ to 0.05 resulted in the highest performance. A lower $\beta$ led to instability in the sampled actions, while higher values negatively impacted the MPC optimization process. Regarding $h_{\text{erc}}$, a value of 5 provided the best results. Extending the horizon did not improve performance, as expert actions taken too far in the future became less informative due to the action constraints.

## A.6 COMPUTATIONAL TIME

In this section, we present the computational time of various baselines and DTWIL during inference. Table 7 reports the average computational time (in seconds) required to generate a single action during inference in HalfCheetah, averaged over 5000 generations. As shown, methods with state-dependent constraints require significantly more time due to the use of the projection function implemented with Gurobi, whereas box constraints, which allow actions to be directly clipped, are much faster.

| | DTWIL | BC+P | GAIL+P | BCO+P | GAIFO+P | OPOLO+P | CFIL-sa+P | CFIL-s+P |
|---|---|---|---|---|---|---|---|---|
| HalfCheetah Box | 0.0002092 | 0.0002164 | 0.0004068 | 0.0003413 | 0.0003860 | 0.0002955 | 0.0010342 | 0.0010611 |
| HalfCheetah HC+O | 0.0337372 | 0.0334898 | 0.0091491 | 0.0104184 | 0.0093199 | 0.0091958 | 0.0099135 | 0.0098245 |

Table 7: Average computation time required to generate a single action during inference, averaged over 5000 trials.

