# OpenReview forum: "Action-Constrained Imitation Learning"
_ICLR.cc/2025/Conference — Submitted to ICLR 2025_

### Official Review · Reviewer_yeRC · 2024-10-25

**Soundness:** 2
**Presentation:** 2
**Contribution:** 2
**Rating:** 5
**Confidence:** 4

**Summary:**

This paper proposes action-constrained imitation learning (ACIL), a new imitation learning algorithm for action-constrained imitators to learn from demonstrations. The authors propose DTWIL to solve this problem by first replacing the original expert demonstrations with a surrogate dataset that follows similar state trajectories and then recasting trajectory alignment as a planning problem and solving it via Model Predictive Control. Through experiments in both navigation and locomotion tasks, they show the effectiveness of proposed method.

**Strengths:**

1. The problem formulation of this paper is novel for tackling action-constrained imitation learning. It tackles the problem by generating demonstration data that adheres to the action constraints.
2. The quantitative experiment results are good.

**Weaknesses:**

1. The motivation requires to be further explained. I am not fully convinced that action-constrained imitation learning is an important problem in a wider range of tasks. The author should give more examples for this point.

2. Also, what is the difference between action-constrained imitation learning and cross-embodiment imitation learning? How do they tackle this problem? The author should elaborate on this more in the introduction/related works sections.

3. From my point of view, the proposed method is based on such an important assumption: the surrogate demonstrations generated by Trajectory Alignment (Section 4.1) must solve the task (or achieve high rewards), so that the imitator can solve the task by doing BC on this data. However, this assumption is not always true for general imitation learning tasks. The author should provide more analysis for this assumption to show what kinds of tasks meet this assumption and what tasks do not. Also, how does the "box constraints" coefficient affect this assumption? For example, if the box constraint of the maze task becomes *action<-0.5 and action >0.5*, can the method also generate good surrogate demonstrations?

4. Why do the IRL and LfO methods (such as GAIL) require a +P operation? Can they directly use the constrained action space for IRL/LfO?

5. Although the proposed method has better interaction sample efficiency than IRL/LfO methods, the author should also show if IRL/LfO methods can solve this task with more interactions, and how many interactions they need to train a successful policy.

6. I doubt if directly using MPC with constrained action space can solve the task with the proposed DTW metric as the (negative) cost function.

7. The stability of the proposed method for other tasks is doubtful. The proposed method requires task-specific operations to make the method effective such as normalization (Section 4.1.1) and time-step alignment (Section 4.2), as well as the $\beta$ hyperparameter.

**Questions:**

See the weakness.

---

> ### Author Response · Authors · 2024-11-21
> **Rebuttal by Authors**
>
> We sincerely thank the reviewer for the thorough and constructive comments. Please find the response to your questions below. To save space, I apologize for reducing your question with "...".
>
> >The motivation requires to be ... examples for this point.
>
> In the second paragraph of the introduction, we have added other practical application examples of ACIL, in addition to the typical applications commonly seen in ACRL. For example, if data is collected using a human to perform tasks, the imitator,
> which may be a robot with hardware limitations, is likely to be unable to replicate the large-scale
> human actions.
>
> >Also, what is the difference ... introduction/related works sections.
>
> Following your suggestion, we have added a section on cross-embodiment imitation learning to the related work paragraph. In this paragraph, we also discussed the differences between ACIL and cross-embodiment imitation learning.
>
> >From my point of view, the proposed method ... generate good surrogate demonstrations?
>
> 1. Thank you for your thoughtful comments. We acknowledge that the assumption—"the surrogate demonstrations generated by Trajectory Alignment (Section 4.1) must solve the task or achieve high rewards"—is indeed a foundational aspect of our method. However, we would like to emphasize that this assumption is relatively mild and is widely adopted in the Learning from Observation (LfO) literature.
> In imitation learning, our goal is to mimic the expert's behavior—in other words, to replicate the expert's trajectory. Therefore, if we successfully align with the expert's trajectory, the surrogate trajectory obtained through alignment will naturally resemble the expert's trajectory and, consequently, achieve a high return. This aligns closely with the core idea of LfO, where solving the task or achieving high return is pursued by mimicking the expert's state trajectory.
>
> 2. We found that the constraint action<-0.5 and action>0.5 have minimal impact on the agent’s performance and would not harm the policy and alignment process. Therefore, we chose action>-0.1 and action<0.1 as our problem.
>
> >Why do the IRL and LfO methods ... action space for IRL/LfO?
>
> 1. Directly using the constrained action space actually has the same effect as the +P operation.
> 2. In real-world tasks, the feasible action space is typically fixed and cannot be changed. However, there are situations where it is necessary to restrict the agent's behavior for safety reasons, as mentioned in the introduction. In such cases, controlling the policy's output through projection is a more preferable approach."
>
> >Although the proposed method ... train a successful policy.
>
> In the appendix, we have included the 500k-step training curves for each baseline method as well as DTWIL. This number of steps is sufficient for all the baselines to demonstrate their capabilities and whether they can effectively train the policy.
>
> >I doubt if directly ... as the (negative) cost function.
>
> Thank you for your insightful comment. We appreciate the opportunity to address this point.
>
> Directly using MPC with a constrained action space and the proposed DTW metric poses challenges in terms of generalizability and practicality. DTW requires a corresponding expert demonstration for alignment, meaning a tailored expert trajectory is needed for each distinct initial state. In real-world scenarios, where the set of starting points is effectively infinite, providing an expert demonstration for every case is infeasible.
>
> Additionally, inference speed is crucial for real-time robotic systems. Using MPC to compute the DTW metric and select optimal actions introduces significant computational overhead, making it unsuitable for time-sensitive tasks.
>
> In contrast, a behavior cloning (BC) policy trained with our method addresses these issues by generalizing across diverse initial states and enabling real-time inference with reduced computational demands, providing a more practical and scalable solution.
>
> >The stability of the proposed method ... as well as the $\beta$ hyperparameter.
>
> Thank you for your observations. We would like to clarify that normalization and time-step alignment are not task-specific components; these designs are applied consistently across all tasks in our experiments. In particular, state normalization is a widely used data preprocessing technique in machine learning and is not tailored to specific environments. Regarding the hyperparameter $\beta_{arc}$​, we acknowledge that it is indeed task-specific and is primarily designed for environments with early termination, such as Hopper. The ARC design stabilizes the MPC-generated actions in Hopper, reducing the likelihood of falling and thereby preventing premature termination. However, the hyperparameter tuning effort for $\beta_{arc}$​​​ is minimal. For instance, a fixed value of $\beta_{arc}$​​=0.05 works effectively across both box-constrained and state-dependent constraint tasks in Hopper, demonstrating its robustness.

---

> > ### Comment · Reviewer_yeRC · 2024-11-25
> >
> > Thanks for your valuable rebuttal. It addresses most of my concerns. I will raise my score to 5.

---

> > > ### Author Response · Authors · 2024-11-27
> > > **Re: Official Comment by Reviewer yeRC**
> > >
> > > We sincerely thank the reviewer for acknowledging our rebuttal and helping us improve our submission.

---

### Official Review · Reviewer_obYj · 2024-10-26

**Soundness:** 3
**Presentation:** 2
**Contribution:** 3
**Rating:** 5
**Confidence:** 3

**Summary:**

The paper proposes a new setting of action-constrained imitation learning where an action-constrained imitator aims to learn from a demonstrative expert with larger action space. The authors show that using behavior cloning to imitation the behavior in the dataset followed by deploying the actions with constraints is insufficient to perform the tasks in their experiments. Accordingly, they propose DTWIL for improving the policy performance in such action constratined settings.

**Strengths:**

- The paper address a new domain of action constrained imitation learning to enable agents to address environmental or physical constraints when performing a task without have the constraints reflected in the expert demonstrations.
- The algorithms uses DTW-based trajectory alignment for modifying the expert trajectories with the agent constraints and perform BC on this modified dataset to obtain policy. The authors also include Actor Regularized Control to improve the alignment’s effectiveness.
- The paper carries out experiments on 3 simulated tasks and provides ablation studies across hyperparameters to justify their design choices.

**Weaknesses:**

- I am a bit confused about the action constrained setting that the paper operates in. For the experiments shown in the paper, the demonstrations are collected with the same agents in the same environments as at test time. So why should additional action constraints be added during deployment when they are not need during demonstration collection? It would be great if the authors could provide a real world example of a scenario needing the introduction of such action constraints during deployment.
- I am a little confused about the algorithm. From what I understand, given some expert demonstrations, these demonstrations are modified through online interactions with the environment. Some questions based on this - (1) In Algorithm 1 Line 3, does each iteration correspond to a trajectory rollout or one step of action in the environment?, (2)  From Eq. 2, it seems like the action sequence optimization is done over a planning horizon. However, Line 6 in Algorithm 1 makes it seem like the whole trajectory is aligned in one go. So does this trajectory alignment involve multiple alignment steps and actions in the environment? (3) In Line 5 of Algorithm 1, its seems like a new forward dynamics model is trained for each iteration on the updated training data. How long does this take? I reckon this might make training slower. (4) In case each iteration corresponds to one environment action step, is a new trajectory sampled at each iteration after having taken action(s) based on a different trajectory in the previous iteration? It would be great if the authors could provide some clarifications regarding this.
- There is a missing reference in line 231.
- In Table 1, for HalfCheetah HC+O, BC+P is the best  performing method but DTWIL is still boldened. Similarly, in Table 3 for DTW-S, Hopper Box-Sync outperforms Hopper Box but Hopper Box is boldened. This is confusing to the reader and beats the purpose of boldening the results.

**Questions:**

It would be great if the authors could address the questions/concerns mentioned in Weaknesses. I am willing to increase my score once these concerns have been addressed.

---

> ### Author Response · Authors · 2024-11-21
> **Rebuttal by Authors**
>
> We sincerely thank the reviewer for the thorough and constructive comments. Please find the response to your questions below.
>
> >I am a bit confused about the action constrained setting that the paper operates in. For the experiments shown in the paper, the demonstrations are collected with the same agents in the same environments as at test time. So why should additional action constraints be added during deployment when they are not need during demonstration collection?
>
> In the second paragraph of the introduction, we have added practical application examples of ACIL, in addition to the typical applications commonly seen in ACRL. In real-world tasks, we will not let the imitator to do behaviors out of his capability if the imitator is a weaker agent because it will hurt itself. Therefore, in the experiments, though the demonstrator and imitator are originally same agents, we add action constraints to the imitator to simulate a weaker imitator.
>
> >I am a little confused about the algorithm. From what I understand, given some expert demonstrations, these demonstrations are modified through online interactions with the environment. Some questions based on this - ...
>
> Sorry for the confusion. Each iteration corresponds to a single trajectory rollout, where the iteration K represents the total number of episodes to be executed. This clarification is now explicitly stated, as highlighted in Algorithm 1 and the newly added Trajectory Alignment pseudocode. Furthermore, it is worth noting that for each episode, PETS updates its dynamics models for approximately 10 epochs — a process that requires minimal time compared to the true computational bottleneck, which lies in the DTW alignment.
>
> >There is a missing reference in line 231.
>
> We have fixed the link. Thanks a lot!
>
> >In Table 1, for HalfCheetah HC+O, BC+P is the best performing method but DTWIL is still boldened. Similarly, in Table 3 for DTW-S, Hopper Box-Sync outperforms Hopper Box but Hopper Box is boldened. This is confusing to the reader and beats the purpose of boldening the results.
>
> We have revised the paper according to the reviewer's suggestions. Additionally, DTW-S represents the DTW distance between surrogate trajectories and expert trajectories. A lower DTW-S value indicates that the surrogate trajectories are more similar to the expert trajectories, so smaller numbers are better.

---

### Official Review · Reviewer_wdKr · 2024-10-28

**Soundness:** 2
**Presentation:** 3
**Contribution:** 2
**Rating:** 5
**Confidence:** 4

**Summary:**

This paper addresses the setting of action-constrained imitation learning, an imitation learning problem where the learned agent must use an action space that is more restricted than the one used by the expert demonstrator. It proposes DTWIL, a method that uses a combination of Model Predictive Control, Dynamic Time Warping, and Behavioral Cloning, to train agents in this setting. The paper shows that the method outperforms several baselines across 3 tasks.

**Strengths:**

The problem setting introduced by the paper is interesting and not widely studied (at least to the best of my knowledge). The paper is well-written and easy to follow. The experiments section compares against a large number of baselines, all of which are reasonable, and the results showcase the effectiveness of the approach.

**Weaknesses:**

There are a few critical weaknesses that should be addressed.

**Limited Experiment Results.** While I appreciate the thoroughness of the results in terms of the number of baselines, the number of tasks shown are small and do not have much variety. For example, Half-Cheetah and Hopper are both locomotion tasks, and relatively simple in complexity (RL approaches can solve these tasks very efficiently from scratch). It would be great to see more settings such as robot manipulation tasks -- there are a wide number of suitable datasets and benchmarks available today (e.g. [robosuite](https://robosuite.ai/), [RLBench](https://sites.google.com/view/rlbench), [ManiSkill](https://www.maniskill.ai/home), and others). It is important to show that the method is general-purpose and easy to apply to many scenarios. Similarly, BC+P seems like a very strong baseline (from Table 1) -- seeing results across more tasks and settings to highlight the value of proposed method would paint a more complete picture.

**Method Limitations.** The ARC (Section 4.2) seems like a hack, and could require per-task tuning, which is undesirable. It is also unclear if it's a good idea to always use blended actions up to a certain timestep, compared to other alternatives for incorporating expert actions, like using the notion of residual additive actions (for example, https://arxiv.org/abs/1812.03201). Including more tasks could help show that one set of parameters works well across multiple settings. It also seems like this method is only suitable for imitating a single specific trajectory, in contrast to typical scenarios where an agent must deal with a variety of initial conditions (such as a robot that needs to manipulate objects that start in diverse configurations on a table from episode to episode). Section 6 mentions "as long as the agent is able to be initialized to the same starting state as the expert" -- this is a severe, and often impractical assumption to make. Finally, it seems like the method implicitly assumes full state observability (e.g. privileged information) compared to partially observed settings (raw sensor data such as images or depth sensing), which also impedes the practicality of the approach.

**Some Writing Issues.** Section 4 does not adequately describe the overall approach of how BC is used with MPC. This needs more details -- I had to figure this out from the Algorithm 1 pseudocode. It also isn't clear why BC is needed versus directly using MPC for control -- some further experiments might help point out the value of using BC.

**Minor Issues.**

- I suggest organizing Related Work further, with a bolded title for each paragraph at least.
- line 231, undefined Figure reference
- Section 5 beginning - "both offline baselines online baselines"

**Questions:**

See "Weaknesses" section.

---

> ### Author Response · Authors · 2024-11-21
> **Rebuttal by Authors**
>
> We sincerely thank you for the constructive comments. Please refer to the new version of the paper and find the response to your questions below.
>
> >Limited Experiment Results
>
> We follow the reviewer's suggestion and conduct experiments in a robot arm environment. We will present the results in the paper once the experiments are complete.
>
> >Method Limitations. The ARC (Section 4.2) seems like a hack, and could require per-task tuning, which is undesirable. It is also unclear if it's a good idea to always use blended actions up to a certain timestep, compared to other alternatives for incorporating expert actions, like using the notion of residual additive actions (for example, https://arxiv.org/abs/1812.03201). Including more tasks could help show that one set of parameters works well across multiple settings.
>
> The reviewer is correct that mixing actions in this manner does not always result in a better-aligned trajectory. In fact, ARC was specifically designed for tasks with early termination, such as Hopper. Without ARC, the actions executed by the MPC agent are not stable enough and often lead to early termination due to a single slightly suboptimal action. To address this, we adopted the ARC approach to make the MPC agent's actions more stable. In environments without early termination, mixing the sampled actions and expert actions may result in a very slight performance drop, but the difference is minimal. The residual additive actions approach the reviewer mentioned can indeed achieve similar effects, but it requires learning an additional residual policy. While the overall framework is similar, training the residual policy might slightly reduce sample efficiency. Therefore, we chose the ARC-based approach.
>
> > Method Limitations. It also seems like this method is only suitable for imitating a single specific trajectory, in contrast to typical scenarios where an agent must deal with a variety of initial conditions (such as a robot that needs to manipulate objects that start in diverse configurations on a table from episode to episode).
>
> We would like to clarify that our method only requires the initial states to be identical during the alignment phase. Once the surrogate demonstration is used to train the behavior cloning (BC) model, the trained policy generalizes across a wide range of initial conditions. This capability allows our method to effectively handle diverse initial states beyond the alignment phase.
>
> To further improve clarity, we have included a statement in Section 5.3 (line 423) explicitly mentioning that the initial points used for evaluation are sampled randomly.
>
> > Method Limitations. Finally, it seems like the method implicitly assumes full state observability (e.g. privileged information) compared to partially observed settings (raw sensor data such as images or depth sensing), which also impedes the practicality of the approach.
>
> Considering that we are taking the first step in ACIL, we would like to focus on the standard setting. Our method is compatible with any MPC algorithm, so this issue can be addressed by replacing the MPC algorithm with one that can handle raw images. For example, TDMPC (https://arxiv.org/abs/2203.04955) is an algorithm capable of tackling partially observable states.
>
> > Some Writing Issues. Section 4 does not adequately describe the overall approach of how BC is used with MPC. This needs more details -- I had to figure this out from the Algorithm 1 pseudocode.
>
> We apologize for the confusion caused by the absence of the Trajectory Alignment pseudocode in the initial submission. To clarify, MPC is employed exclusively during the Trajectory Alignment phase, where it generates hundreds of candidate rollouts for alignment with the expert trajectories. This process is distinct from the Behavioral Cloning (BC) component, which is used at the end of the pipeline (Algorithm 2 in the revised version) to train a policy using all aligned surrogate trajectories.
>
> > Some Writing Issues. It also isn't clear why BC is needed versus directly using MPC for control -- some further experiments might help point out the value of using BC.
>
> The inclusion of the BC model provides a significant advantage: it enables our method to generalize effectively to unseen initializations and rollouts that were not present in the expert dataset. In contrast, directly using MPC for control inherently relies on having corresponding expert trajectories for alignment, limiting its ability to generalize.
>
> > I suggest organizing Related Work further, with a bolded title for each paragraph at least.
>
> Following the reviewer's suggestion, we have added bolded titles to each paragraph
> and streamlined some of the content.
>
> > line 231, undefined Figure reference
>
> We have fixed the link. Thanks a lot!
>
> > Section 5 beginning - "both offline baselines online baselines"
>
> We have revised this line to "both offline baselines and online baselines". Please check it out (line 358).

---

> > ### Comment · Reviewer_wdKr · 2024-11-27
> > **Response**
> >
> > Thanks for the clarifications and revisions!
> >
> > I still do not understand how the final agent can handle different initializations if MPC only generates demonstration data using a single initial state. Can the authors clarify?

---

> ### Author Response · Authors · 2024-11-27
> **Re: Response**
>
> We greatly appreciate the reviewer's acknowledgment of our rebuttal and suggestions for improvement. We are happy to provide further details.
>
> The MPC is, in fact, reset to the initial state of each expert trajectory, which can vary between trajectories. Its role is solely to generate surrogate expert data that aligns with the original expert trajectory. Once this surrogate expert data is obtained, we use BC to train the agent's policy, which can handle different initializations. Therefore, DTWIL's generalization capability is equivalent to that of BC. As long as the diversity of starting points in the expert data is sufficient for BC, the diversity of starting points in the surrogate expert data will also be sufficient for DTWIL.

---

### Official Review · Reviewer_Viqr · 2024-10-31

**Soundness:** 3
**Presentation:** 2
**Contribution:** 3
**Rating:** 5
**Confidence:** 3

**Summary:**

This paper proposes a novel problem setting called Action-Constrained Imitation Learning (ACIL), where an imitation learner's action space is a subset of an expert's action space. The paper proposes DTWIL, which (1) generates feasible trajectory for the imitator by using Model Predictive Control (MPC) with Dynamic Time Warping (DTW) as its objective, and (2) performs behavior cloning from the generated trajectories. Experimental study showed that, existing imitation learning (IL) methods with a naive projection approach suffer from low performances in ACIL settings, and DTWIL outperforms these methods.

**Strengths:**

- The proposed paradigm, ACIL, is an important research direction. It must have many potential applications and a large group of potential audiences (originality, significance).
- The high level idea of DTWIL seems reasonable and novel, and the experimental study showed that it indeed performs better than baselines in ACIL problems (originality, quality, significance).

**Weaknesses:**

Besides the strengths above, the presentation quality of the paper is not good in general. The followings are my concerns.

### Major concerns
- The exposition of DTWIL is not self-completed and clear enough. For example,
  - At L.231, there is a statement "The pseudo code for trajectory alignment is presented in __??__" and the pseudo code for __trajectory alignment__ is missing in the paper.
  - The definition of DTW distance in Eq. (2) is not provided, though its high level idea is stated in page 4.
  - Since trajectory alignment algorithm is missing, it is not clear how the action constraints are handled practically. I conjecture that Eq. (2) is solved by a constrained optimization problem.
  - The definition of $\bar{S}(A,f_\theta)$ is not concrete. Does this sequence start from $\bar{s}_{t_{\rm pg}}^e$?
  - How $t_{\rm pg}$ is updated in practice? Figure 3 explains only discrete cases. For continuous spaces, I suppose that we need to compute the distance of states by some metric and determine by a threshold, which must be an additional hyper parameter to be tuned.

- I am not convinced of the validity of Actor Regularized Control (ARC). In my understanding, the expert actions before projection, $a^e$, are sampled from the dataset. On the other hand, $a^{\rm sampled}$ are computed for states in the generated trajectory. Therefore, the states for which $a^e$ and $a^{\rm sampled}$ are sampled are different by construction. Mixing these different-state-dependent actions seems not valid.

- Ablation study is not comprehensive enough. Since the paper describes the importance of excluding the final expert state in Figure 3, the reader may expect its experimental impact.

- The time complexity of the naive DTW algorithm is O(NM), where N and M are the lengths of the two input sequences. I suppose that DTWIL has a drawback in the computational complexity compared to the baselines, which might hinder the applicability of DTWIL to real world applications. I think that it is necessary to compare the computational time with baseline methods.


### Minor concerns
- For Maze2d, state-dependent constraint is not stated.
- Section 5.6 and Appendix A.3 look exactly the same.


## After reading author responses and revised manuscripts
- As of 11/25: Thank you for revising the paper. The presentations are improve largely. I raise the score from 3 to 5.
- As of 12/03: Thank you for clarifications. I raise the score from 5 to 6.

**Questions:**

### Questions
- How the action constraints are handled practically in DTWIL? Is Eq. (2) solved by a constrained optimization problem?
- How ARC in Eq. 3 is justified?

### Suggestion
- Please address my concerns, especially in the exposition of DTWIL, and update the manuscript in Discussion Stage. In the current form of the paper, the contributions are not clear. Considering the strengths of this paper, I am happy to re-evaluate after revision.

---

> ### Author Response · Authors · 2024-11-21
> **Rebuttal by Authors**
>
> We sincerely thank the reviewer for the constructive comments. Please refer to the new version of the paper and find the response to your questions below.
> > At L.231, there is a statement "The pseudo code for trajectory alignment is presented in ??" and the pseudo code for trajectory alignment is missing in the paper.
>
> We have fixed the link. Thanks a lot!
>
> >The definition of DTW distance in Eq. (2) is not provided, though its high level idea is stated in page 4.
>
> We have added an explanation of the meaning of DTW distance in the Preliminaries section (line 188) in the new version.
>
> >Since trajectory alignment algorithm is missing, it is not clear how the action constraints are handled practically.
>
> We have fixed the link to Trajectory Alignment. Additionally, right after Eq. (2), starting from line 262, we added some explanations, mainly about that we solve this optimization problem approximately, and that we use rejection sampling to ensure actions to conform to the constraints. We hope it is clearer now.
>
> >The definition of  $\bar S (A, f_{\theta})$ is not concrete. Does this sequence start from $ \bar{s}^e_{t_{pg}} $?
>
> We thank the reviewer very much. We removed this ambiguous notation, and now using $s_{t:t+H}$ instead to represent the same meaning.
>
> >How  $t_{pg}$ is updated in practice? Figure 3 explains only discrete cases. For continuous spaces, I suppose that we need to compute the distance of states by some metric and determine by a threshold, which must be an additional hyper parameter to be tuned.
>
> The table in this figure does not represent environments with discrete state spaces, such as grid worlds. Instead, it is a memoization cost matrix for dynamic programming for DTW computation. It illustrates the warping path that needs to be identified within the DTW matrix during the computation of the distance between two trajectories.
>
> >I am not convinced of the validity of Actor Regularized Control (ARC). In my understanding, the expert actions before projection, $a^e$, are sampled from the dataset. On the other hand,  $a^{sampled}$ are computed for states in the generated trajectory. Therefore, the states for which $a^e$ and $a^{sampled}$ are sampled are different by construction. Mixing these different-state-dependent actions seems not valid.
>
> Before mixing $a^e$ ​ and $a^{sampled}$, we first project $a^e$​ onto the feasible set that $a^{sampled}$ is required to satisfy. Thus, the mixture of two actions that satisfy the same state-dependent action constraint will also satisfy this state-dependent action constraint.
>
> >Ablation study is not comprehensive enough. Since the paper describes the importance of excluding the final expert state in Figure 3, the reader may expect its experimental impact.
>
> We thank the reviewer for your suggestion. We have done this additional ablation study and showed the results at section 5.5. For Maze2d-Medium, excluding the final expert state improves the success rate by about 5%.
>
> >The time complexity of the naive DTW algorithm is O(NM), where N and M are the lengths of the two input sequences. I suppose that DTWIL has a drawback in the computational complexity compared to the baselines, which might hinder the applicability of DTWIL to real world applications. I think that it is necessary to compare the computational time with baseline methods.
>
> Though the training time is substantial, there are several strategies available to mitigate this during the training phase. Moreover, in real-time applications, inference time typically takes precedence over training duration. Thus, this aspect would not be a critical limitation, as our method benefits from a relatively short testing time due to the use of a behavioral cloning (BC) model for inference.
>
> >For Maze2d, state-dependent constraint is not stated.
>
> We thank the reviewer for your suggestion. We will include the results for the state-dependent Maze2d environment in the experiments chapter. As this was not part of our initial experimental plan, generating these results may require a few additional days. Currently, the experiments are still in progress.
>
> >Section 5.6 and Appendix A.3 look exactly the same.
>
> They are fine in this new version. Please check it out.
>
> **Reference**
>
> [1] Sakoe Hiroaki and Seibi Chiba. Dynamic programming algorithm optimization for spoken word
> recognition. IEEE Transactions on Acoustics, Speech, and Signal Processing, 1978.
>
> [2] Justin Fu, Aviral Kumar, Ofir Nachum, George Tucker, and Sergey Levine. D4rl: Datasets for deep
> data-driven reinforcement learning. arXiv:2004.07219, 2020.

---

> ### Comment · Reviewer_Viqr · 2024-11-25
> **Thank you for revising; follow-up comments and questions**
>
> I appreciate the authors' extensive effort to revise the paper. I acknowledge that the presentation of the paper improved largely. I raise the score from 3 to 5. However, I believe there are more room of improvements as listed below.
> - I believe that the definition of DTW distance must be explicitly provided. The current form of the paper is not self-contained. It still lack the explanation related to the cost matrix, which makes hard the reader to understand what Figure 3 means.
> - I understand that, the authors' claim that ERC is valid because "the feasible set that $a^{\rm sampled}$ is required to satisfy", which is __state dependent__. However, __L321__ simply states that "onto the constrained action space". I believe this state-dependence must be clearly denoted.
> - On computational complexity compared to the baselines: could you explicitly show the computational time required for DTWIL and allowed time in real-time applications? It must help the readers to understand the soundness of the proposed method.

---

> > ### Author Response · Authors · 2024-11-27
> > **Re: Thank you for revising; follow-up comments and questions**
> >
> > We sincerely thank the reviewer for acknowledging our rebuttal and helping us improve our submission. We have revised the paper as suggested by the reviewer.
> >
> > > I believe that the definition of DTW distance must be explicitly provided. The current form of the paper is not self-contained. It still lack the explanation related to the cost matrix, which makes hard the reader to understand what Figure 3 means.
> >
> > We followed the reviewer's suggestion and have revised the Preliminaries section to enhance readability by incorporating additional explanations in lines 186–198.
> >
> > > I understand that, the authors' claim that ERC is valid because "the feasible set that $a^{sampled}$ is required to satisfy", which is state dependent. However, L321 simply states that "onto the constrained action space". I believe this state-dependence must be clearly denoted.
> >
> > We sincerely thank the reviewer for their suggestion. In response, we have revised the notation for constrained action spaces and the projection function. Additionally, we have provided further explanation to enhance clarity. The updated content can be found in lines 344–356 of the revised version of the paper.
> >
> > > On computational complexity compared to the baselines: could you explicitly show the computational time required for DTWIL and allowed time in real-time applications? It must help the readers to understand the soundness of the proposed method.
> >
> > We have included a related table in Appendix 6 for further reference.

---

> ### Comment · Reviewer_Viqr · 2024-12-03
> **Thank you for clarifications.**
>
> Thank you for the clarifications. My original concerns are mostly resolved. I raise the score to 6.
>
> Minor issues:
> - $H$, $h_{erc}$ and $P$ are not used in Algorithm 1; I supposes these values are fed into `Trajectory Alignment` as well as $\tau^{i}$.
> - Eq.(3): ${\rm Proj}(a^e_{t_{pg}+h}|s_h)$ -> ${\rm Proj}(a^e_{t_{pg}+h}|\mathcal{C}(s_h))$

---

### Meta-Review · Area_Chair_8K46 · 2024-12-18

**Metareview:**

This paper studies a novel problem formulated as action-constrained imitation learning where the imitation learning agent has access to only a subset of the demonstrator's actions. The proposed learning algorithm leverages dynamic time warping to measure discrepancies between trajectories, generates surrogate data with actions from the learner's action space using model predictive control, and then train a behavior cloning policy with the surrogate data.

The reviewers agree the problem studied in this paper is novel and can have a high impact on robotics applications, but raised concerns about the assumption the proposed algorithm makes, as well as the lack of sufficient evaluation on benchmark tasks and comparison with state-of-the-art baselines. Reviewers suggest that incorporating additional empirical results and testing in realistic task settings would strengthen the paper.

In summary, while the paper presents a novel problem and a plausible solution, further empirical validation and broader evaluations are necessary to fully establish its efficacy and applicability.

**Additional Comments On Reviewer Discussion:**

The authors addressed reviewers' initial concerns on clarity and presentation of the work during the rebuttal and discussion period. Reviewers acknowledged that the revised version has been greatly improved but the technical contents can be further improved.

---

### Decision · Program_Chairs · 2025-01-22

Reject